# Single-Cell RNA Sequencing with Spatial Transcriptomics of Cancer Tissues

**DOI:** 10.3390/ijms23063042

**Published:** 2022-03-11

**Authors:** Rashid Ahmed, Tariq Zaman, Farhan Chowdhury, Fatima Mraiche, Muhammad Tariq, Irfan S. Ahmad, Anwarul Hasan

**Affiliations:** 1Department of Mechanical and Industrial Engineering, College of Engineering, Qatar University, Doha 2713, Qatar; 2Biomedical Research Centre, Qatar University, Doha 2713, Qatar; 3Department of Biotechnology, Faculty of Natural and Applied Sciences, Mirpur University of Science and Technology, Mirpur 10250 AJK, Pakistan; tariq.awan@must.edu.pk; 4Nick Holonyak Jr. Micro and Nanotechnology Laboratory, University of Illinois at Urbana Champaign, Urbana, IL 61801, USA; isahmad@illinois.edu; 5College of Human Medicine, Michigan State University, Grand Rapids, MI 49503, USA; tariqzaman.ak@gmail.com; 6Department of Mechanical Engineering and Energy Processes, Southern Illinois University Carbondale, Carbondale, IL 62901, USA; farhan.chowdhury@siu.edu; 7Department of Pharmaceutical Sciences, College of Pharmacy, QU Health, Qatar University, Doha 2713, Qatar; fatima.mraiche@qu.edu.qa; 8Department of Agricultural and Biological Engineering, University of Illinois at Urbana Champaign, Urbana, IL 61801, USA; 9Carle Illinois College of Medicine, University of Illinois at Urbana Champaign, Urbana, IL 61801, USA

**Keywords:** intratumor heterogeneity, single-cell RNA sequencing techniques, spatial transcriptomics

## Abstract

Single-cell RNA sequencing (RNA-seq) techniques can perform analysis of transcriptome at the single-cell level and possess an unprecedented potential for exploring signatures involved in tumor development and progression. These techniques can perform sequence analysis of transcripts with a better resolution that could increase understanding of the cellular diversity found in the tumor microenvironment and how the cells interact with each other in complex heterogeneous cancerous tissues. Identifying the changes occurring in the genome and transcriptome in the spatial context is considered to increase knowledge of molecular factors fueling cancers. It may help develop better monitoring strategies and innovative approaches for cancer treatment. Recently, there has been a growing trend in the integration of RNA-seq techniques with contemporary omics technologies to study the tumor microenvironment. There has been a realization that this area of research has a huge scope of application in translational research. This review article presents an overview of various types of single-cell RNA-seq techniques used currently for analysis of cancer tissues, their pros and cons in bulk profiling of transcriptome, and recent advances in the techniques in exploring heterogeneity of various types of cancer tissues. Furthermore, we have highlighted the integration of single-cell RNA-seq techniques with other omics technologies for analysis of transcriptome in their spatial context, which is considered to revolutionize the understanding of tumor microenvironment.

## 1. Introduction

Tumor heterogeneity is the most complex contributor to misdiagnosis and is often associated with the failure of identifying its margins in different tissues. There is a dire need for the development of new approaches that can precisely detect molecular mechanisms fueling tumor heterogeneity and better help diagnose cancers in clinical practices. Molecular approaches developed so far provide limited information about tumor cells and their microenvironment as these approaches can perform bulk transcriptome analysis without addressing their spatial context [1]. In the case of heterogeneous cancer, imaging and molecular analysis face the challenges of identification of aggressive clones in their spatial context leading to poor efficacy of therapeutic interventions [2]. Single-cell RNA-seq can perform sequence analysis of the whole transcriptome of cancer tissue sections to understand the complex landscape of the tumor microenvironment [3]. Homogenization of tissue samples during single-cell RNA seq analysis (before sequencing) usually disrupts the spatial information of cells and thus leads to failure in the understanding of organizations of cells and their interaction with each other in the native tissue landscape. The issue of loss of spatial information about RNA analytes has been resolved by combining single-cell RNA-seq with spatial transcriptomics. The combination of these approaches helped profile RNA expression in their native site and improved understanding of the factors that determine morphology, genotype, and microenvironment of the cells, which can help develop precise diagnosis and effective treatment strategies [4]. 

Analysis of RNA expression in their spatial context provides important information about cellular heterogeneity of tissues, tumors, and immune cells as far as the application of spatial transcriptomics is concerned. This also helps to decipher subcellular localization and expression of RNA in various conditions, which leads to a better understanding of the architecture of tissues [5]. Spatial transcriptomics approaches provide an opportunity to understand the functioning of individual cells in complex multicellular organisms by knowing their physical location in tissue sections. In the past few years, there have been tremendous efforts to map the transcriptome of cells in their spatial context, and in this direction different omics approaches have been successfully applied for the analysis of cells in tumor tissue [6]. It has been proved that these methods are complementary to the single-cell RNA-seq and can be integrated for the mapping of RNA analytes within tissue samples. For instance, it is found that when in situ hybridization (ISH) is combined with the single-cell RNA-seq technique we obtain information of transcriptional heterogeneity existing among various types of cells in tissue architecture. However, ISH methods face certain challenges for the molecular analysis of RNA as there is no reliable ISH method for the analysis of solid tumors that display varied gene expression patterns due to variable tissue architecture. In addition, ISH techniques have limited throughput and are only applicable to a small subset of transcripts only. However, recent developments in omics technologies such as spatial transcriptomics have largely resolved the limitations of ISH methods, especially issues of limited throughput and accuracy. Furthermore, these platforms can profile the expression patterns of RNA analytes within cells without tissue homogenization [7]. Recent progress in spatial transcriptomics technologies has made it possible to visualize the transcriptome of 100–200 cells [8]. Thus, a combination of single-cell RNA seq and spatial transcriptomics provides an unbiased and in-depth analysis of heterogeneous tissue samples, especially when spatial visualization of RNA transcripts is required. 

This review article describes potential applications of the emerging single-cell RNA-seq techniques for the analysis of unbiased RNA sequencing within the heterogeneous tumor and highlights the importance of single-cell RNA-seq techniques in terms of their accuracy, sensitivity, reliability, and resolution for exploring and understanding the microenvironment. The review article also provides an in-depth knowledge of the integration of single-cell RNA-seq with data generated by omics technologies towards a comprehensive analysis of intratumor heterogeneity. It is expected that the literature presented here will help researchers expand their work for exploring splicing and post-transcriptional modifications of RNA methylation, without compromising the spatial context of the cancer cell using single-cell RNA-seq techniques. 

## 2. History of Single-Cell RNA-Seq Techniques

The journey of single-cell RNA-seq started in 2009 when this technique was used for the first time for the analysis of mouse blastomere dividing at the four^-^cell stage [9]. In the same year, the first-ever single-cell transcriptome was analyzed with this technique, and work was published by using the next-generation sequencing platform to assess the characteristics of the cells from early developmental stages. James Eberwine et al. [10] and Iscove along with their colleagues [11] were pioneers for sequencing the entire transcriptome at the single-cell level. The first multiplex single-cell RNA-seq library was created by Islam et al. using mouse embryos [12]. Up till 2012, single-cell RNA-seq techniques were able to explore unprecedented details in gene expression analyses. However, their efficient application to single-cell transcripts analysis was challenged by the small starting amounts of RNA. This problem was overcome by the development of cell expression by linear amplification and sequencing (CEL-Seq), a method for overcoming this limitation by barcoding and pooling the samples [13]. The first multiplex single-cell RNA-seq methodology was commercialized in 2014, which greatly helped decrease the time and labor required for isolation and library preparation of single cells [14]. In the following year, the drop-seq technique was developed for barcoding the RNA from thousands of individual cells [15], which showed a surprisingly low noise profile during its application for transcripts analysis. By the end of 2016, Tirosh et al. explored the distinct genotypic and phenotypic states of melanoma tumors by application of single-cell RNA-seq to 4645 single cells isolated from 19 patients [16]. Furthermore, further advancement took place when single-cell RNA-seq was carried out on clustered cell populations in the murine epidermis, which resulted in the identification of 25 different types of cells from the tissue samples [17]. A study performed by Shalek and Benson showed that single-cell RNA-seq plays a key role in understanding the molecular pathways associated with disease development during personalized medications [18]. In 2017, the seq-well was introduced, which served as the simplest, but affordable and portable, single-cell library preparation platform for analysis of thousands of human macrophages exposed to *Mycobacterium tuberculosis* [19]. Another technique named transcriptional landscape during cytomegalovirus latency was introduced in 2018 by using the single-cell RNA-seq technique [20]. Furthermore, a technique named single-cell optical phenotyping and expression sequencing (SCOPE-Seq) was introduced, which was able to perform single-cell RNA seq with live-cell imaging [21]. The latest research in this field shows the innovation of the RNA-Seq toolbox for human tissues which have currently analyzed 216,490 cells along with their nuclei and have detected eight tumor types in tissue samples [22]. Recently there are several droplet-based platforms such as Chromium from 10x Genomics, InDrop from 1CellBio, ddSEQ from Bio-Rad Laboratories, and μEncapsulator from Dolomite Bio/Blacktrace Holdings which provide reagents on a commercial basis for doing the single-cell RNA-seq for wet labs. The uniqueness of droplet-based instruments is that there are individual partitions in these instruments which contain all the necessary reagents for cell lysis, reverse transcription, and molecular tagging. Thus, cells can be encapsulated inside these partitions and there is no need for isolation of cells through flow-cytometry or microdissection [15,23,24]. In this way, thousands of cells can be analyzed by single-cell RNA-seq. However, there should be a specialized hardware platform provided to the researchers for performing single-cell RNA seq with droplet-based methods for the first time. This can be seen in a timeline of discovery and innovations of single-cell RNA sequencing techniques as shown in Figure 1. 

## 3. Methods of Single-Cell RNA-Seq Techniques

There have been tremendous efforts carried out for the development of single-cell RNA seq techniques in wet-lab, bio-informatic or computational tools in recent years, resulting in the general methodology as presented in the methodological pipeline (Figure 2). Among these techniques, the most important step is the isolation of single cells from tissue samples. Isolated cells are lysed with specific reagents for the extraction of target RNA molecules and processed after the purification process. We would like to mention here that RNA molecules are polyadenylated with poly(T) primers to prevent the presence of ribosomal RNA in a reaction as processing (analysis) of non-polyadenylated mRNA is quite challenging and requires specific protocols [25,26]. Total RNA molecules isolated from cells are used to synthesize cDNA molecules by reverse transcriptase reaction. In addition, to reverse transcription primer sets, specific adopter sequences (unique molecular identifiers) are added for the detection of RNA molecules by NGS platforms [27]. Synthesized cDNA molecules are amplified using PCR or in vitro transcription followed by another round of reverse transcription or amplification by nucleotide barcode-tagging [28]. Then, amplified and tagged cDNA from every cell is pooled and sequenced by NGS, using library preparation techniques, sequencing platforms, and genomic-alignment tools similar to those used for bulk samples [29]. The steps used for carrying out the RNA sequencing from single-cell levels obtained from cancer tissues are shown in Figure 2. To date, there are several methods used for sequencing transcriptome of single-cell level, but the most important techniques that have been commonly used in single-cell RNA-seq include SCRB-seq, CEL-seq2, MARS-seq, Drop-seq, Smart-seq1, Smart-seq2, and 10× Genomics. The Smart-seq2 could detect maximum transcripts in an individual cell while CEL-seq2, MARS-seq, Drop-seq, and SCRB-seq can quantify mRNA with minor noise using unique molecular identifiers (UMIs). However, the most cost-effective technique is the Drop-seq for RNA profiling in cases where analysis involves a higher number of cells, but for fewer cells, SCRB-seq, MARS-seq, and Smart-seq2 approaches are more useful and more effective for profiling transcriptome. 

### 3.1. Cell Expression by Linear Amplification and Sequencing (CEL-Seq)

Single-cell RNA-seq introduced by Tang and his colleagues involved the use of polyT sequences for the analysis of mRNA from tissue samples [9]. Initially, it was found suitable for the analysis of RNA as this technique can profile 75% more RNA analytes compared with microarray techniques, but it required a higher concentration of input RNA for the seq processing. To solve this issue, Hashimoshony et al. used a CEL-Seq method that involved barcoding and the pooling of RNA from tissue samples [13]. The method involved barcoding, 3′ end tagging, and analysis of transcripts. CEL-seq has some interesting features which were missing in other methods. For instance, it has higher strand specificity (with more than 98% of exonic reads coming from the sense strand) and high barcoding efficiency (>96%) [13]. Moreover, because only the RNA fragments that are the closest to the poly(A) tail are selected, the estimation of expression levels is much easier than with full-length RNA-seq methods.

CEL-seq has higher 3′-bias and low sensitivity for lowly expressed transcripts. A transcript that has 5 RNA copies per cell, CEL-seq has only a 50% chance of identifying these transcripts [13]. While sequencing the 3′-terminal portion of each transcript is certainly sufficient for determining cell identities in a heterogeneous population, it is not suitable for obtaining detailed information of the rent splice isoforms and their relative abundances in the cell.

The challenges described in the CEL-seq were overcome when Hashimoshony et al. developed an improved version of CEL-seq named CEL-seq2 in 2016. This method has a much higher sensitivity for the detection of RNA transcripts from tissue samples and needs reduced costs and time for the analysis of tissue samples [30]. Figure 3 shows different types of RNA-seq techniques used for the analysis of RNA from cancer tissues.

### 3.2. Single-Cell RNA Barcoding and Sequencing (SCRB-Seq)

SCRB-seq was developed as it can perform analysis of RNA analytes from a low input concentration. This method involves the use of barcoding and sequencing of single-cell RNA by incorporating unique molecular identifiers (UMIs) in the reaction, which has considerably reduced the amplification bias of target mRNA [31]. Later, an advanced form of SCRB-seq was developed in the form of molecular crowding SCRB-seq (mcSCRB-seq) that involved the isolation of single cells by FACS sorter in multi-well plates which contained reaction mix. The reaction used in the mcSCRB-seq technique also used PEG 8000 (7.5%) for the reverse transcription process and for switching the template for molecular crowding conditions [32]. Advantages of SCRB-seq included being high-throughput, cost-efficient, and having single-cell transcriptome profiling capacity and higher gene-detection capacity with improved sensitivity as compared with other well-known single-cell RNA-seq techniques. The disadvantage of this method is that template-switching reverse transcription is highly biased for full-length mRNA [33].

### 3.3. Switching Mechanism at the End of the 5′-End of the RNA Transcript Sequencing (Smart-Seq)

One of the state-of-the-art next-generation sequencing methods is SMART-seq. Although it is more suitable for sequencing small transcripts, it can also be used to read full-length genes to show complex structural variation existing in the DNA samples, i.e., indicating where copy number variation has occurred based on the reference sequence [34]. In this context, the work of Ramskold et al. received pronounced attention for their aim to improve the analysis of samples involving the number and size of transcripts. The technique permitted analysis of forty percent of transcripts from 10 pg of RNA samples, which is equal to an amount of RNA present in a single cell. The technique used by Ramskold et al. [34] involved the use of moloney murine leukemia virus for the preparation of cDNA from RNA from a single cell level. This method produces transcripts intact at 5′ ends and, therefore, there is no need for the second strand synthesis [35]. These techniques are powerful enough to resolve complex RNA splicing patterns from chromosomal DNA libraries, as a single long read might consist of the entire transcript from one end to another. Smart-seq is known to offer many advantages such as needing a very minute concentration of input RNA (50 pg). It can perform analysis with better coverage across RNA transcripts as well as show a high potential of mapping the transcripts. However, its demerits include lack of early multiplexing capacity, lack of strand-specificity, absence of transcript length bias when transcript length is over 4 Kb, and requiring higher input RNA for the amplification. Moreover, the purification step might result in the loss of material due to strand-invasion bias. Although Smart-seq dramatically improved the coverage of the transcriptome and had much higher sensitivity compared with the Tang method, it bore some important limitations. A lower read coverage toward the 5-end of the transcripts, especially long several kilobases, was still quite pronounced. Moreover, in the final sequencing library, an under-representation of transcripts with a high GC content was observed, presumably an effect of the complex secondary structure of the RNA that the DNA polymerase could not overcome during the PCR [36]. Most importantly, having to buy an expensive commercial kit made library preparation prohibitive for research groups on a tight budget and planning to sequence hundreds or thousands of cells. To address all these issues and improve the existing method a fairly large selection of buffers, additives, and enzymes, as well as reaction conditions, were tested. Hundreds of experiments resulted in a dramatically better protocol that, perhaps without too much creativity, was named Smart-seq [36,37,38]. From experiments on different cell lines, the authors observed a substantial increase in the ability to detect gene expression and a lower technical variation for low- and medium-abundance transcripts compared with any other full-length single-cell method. The improved sensitivity led to the detection of a couple of thousand genes more than with Smart-seq, especially those with a high GC content [36]. In addition, an advanced version of smart-seq named as smart-seq2 was developed which was able to analyze the transcriptome of single-cell for full-length chromosomal DNA and sequencing library by using standard reagents commonly used in molecular biology labs [39,40].

### 3.4. Drop-Sequencing (Drop-Seq)

Drop-seq is a cost-friendly technique compared with other sequencing techniques for the analysis of transcriptome at the single-cell level, and the method relies on the use of encapsulation of single cells with DNA barcoded microbeads [23]. This also allows maintaining the information on the origin of the transcript at the cellular level. The molecular barcoding beads are used to distinguish the cell of origin of each mRNA.

Drop-seq is a cost-effective technique as USD 0.7 is needed for one cell analysis and library preparation time is very short (10,000 per day). The identification of mRNA strands is quite easy with unique molecular and cell barcodes used in this technique. This method results in the creation of a high amount of cDNA using reverse transcription with template-switching PCR. The merits of Drop-seq include evaluation of single-cell sequences of similar patterns, identification of gene-specific mRNA strands via single molecular and cell barcodes, high amount of reads from single cells, cost-effectiveness, and fast library prep (10,000 cells per day). However, it requires a specialized custom microfluidic device for the separation of droplets, has low gene sensitivity per cell compared with other single-cell RNA-seq techniques, and is restricted to only the mRNA transcripts.

### 3.5. Massively Parallel RNA Single-Cell Sequencing Framework (MARS-Seq)

MARS-seq (automated massively parallel RNA sequencing framework) was developed for sampling thousands of in vivo cells that used RNA sequencing multiplex by sustaining the control over biases during amplification and labeling errors. For exploring this new technique, RNA from over 4000 mouse spleen single cells was sequenced by focusing on the heterogeneous cell population having a high level of expression of surface marker CD11c to differentiate between the diversity of spleen cells and cells of ductal carcinoma. A new framework was developed from single-cell transcriptional states of tissues to break down the complex functions. MARS-seq can be applied easily to tissues/organs for revealing detailed genome-wide transcriptional profiling in normal and diseased states, thus proving this technique to play a vital role to study the biological functions of cells in vivo [23]. MARS-seq involves the use of randomized molecular tags to initially label poly-A tailed RNA molecules, followed by pooling labeled samples and performing two rounds of amplification, generating sequencing-ready material. 

Advantages of MARS-seq include in vivo sampling, high throughput transcriptional profiling, and three barcode levels (cellular, molecular, and plate-level tags), which help in vigorous multiplexing skills and are cost-effective. However, disadvantages are three biases in the purification step and removal of strand-specific information during fragmentation. Table 1 shows a precise comparison of different single-cell RNA-seq techniques for the analysis of transcriptome of a cell.

### 3.6. 10x Genomics Single-Cell RNA-Seq

There have been considerable contributions from 10x Genomics in single-cell RNA seq analysis which has led to the identification of various cell types (subpopulations) found in heterogenous cancer tissues. For instance, the GemCode™ Technology from 10x Genomics, which can analyze the transcriptome of a huge number of cell populations (thousands of cells) by combining a microfluidic platform with molecular barcoding and then the analysis of cells transcriptome at single-cell resolution using custom bioinformatics software [43]. There is another useful technology developed by 10x Genomics, named Chromium™ Single Cell RNA sequencing, for the analysis of the transcriptome of single-cell [42]. This technology performs the analysis of single-cell RNA seq by encapsulating reaction reagents, single cells, and barcoded oligonucleotides coated over a single Gel Bead into nanoliter-sized GEMs (Gel Bead in the emulsion). After encapsulation of reagents needed for reaction inside the GEM, lysis of a single cell take place and polyadenylated mRNA undergo barcoded reverse transcription. This whole process creates high-quality next-generation sequencing libraries of the target transcripts in a single bulk reaction. mRNA libraries are analyzed with the Chromium™ Software Suite to visualize the gene expression at a single-cell level. Chromium™ Single Cell RNA sequencing has high-throughput and thus is used for measuring gene expression at single-cell to profile individual cell types in tissues. This system is a complete solution for the profiling of RNA expression in a single cell and offers the best solutions for identifying subpopulations of rare cell types in heterogeneous tissues. It is more efficient compared with droplet systems as it can process more than 80,000 cells in less than 10 min, so it offers a wide dynamic range as compared with droplet systems.

## 4. Spatial Transcriptomics

Spatial transcriptome analysis is one of the breakthroughs in the field of medical biotechnology as this can map the analytes such as RNA information in their physical location in tissue sections [44]. This term was first used by Doyle et al. in 2000, termed spatial genomics [45]. This approach was adopted by Stahl et al. in 2016 and applied for the analysis of mRNA analytes with improved resolution and higher sensitivity [7]. After the work of Stahl et al. in spatial mapping of RNA, analytes are performed with different approaches such as in situ sequencing, fluorescent in situ hybridization approaches, in situ capture techniques, and in silico methods [5]. However, spatial transcriptomics of RNA analytes is divided into two broad categories [46,47], (1) next-generation sequencing (NGS)—comprising positional analysis of RNA transcripts before next-generation sequencing and (2) imaging-based techniques including in situ sequencing-based methods, which involve amplification of RNA and their sequencing in a tissue sample and in situ hybridization-based approaches. The analysis of transcripts is carried out using imaging probes that hybridized sequentially in the tissue [48,49,50].

### 4.1. Next-Generation Sequencing (NGS)-Based Approaches

Single-cell RNA-seq techniques have been the basis for performing the transcriptome analysis by NGS-based approaches as these methods involve the incorporation of a spatial barcode before library preparation [51]. The First NGS-based study was reported by Stahl et al. in 2016 for spatial transcriptomics for tissue sections [7]. It is worth mentioning here that 10x genomics has played a major contribution in the development of spatial analysis of RNA entities taken from biological tissues using microarray approaches and barcoding techniques. Spatial transcriptomics is capable of analyzing whole transcriptome data across a tissue section and has the potential to visualize any number of genes within a tissue section [5]. The technique of spatial transcriptomics involved the capturing of RNA analytes on spatially barcoded microarray slides before proceeding to the reverse transcription step. This ensured the mapping of each RNA molecule to its original position in the tissue sample using the unique positional molecular barcode [51]. This technique was tested on the tissue samples from mouse olfactory bulbs [7] and afterward employed on several other tissue samples [52,53,54]. This technique was adopted by 10x Genomics and the company has recently introduced a new technology called Visium which has improved resolution and higher sensitivity.

Another form of NGS-based platform is Slide-Seq, which performs analysis of transcripts by covering the glass slides with randomly barcoded beads to capture mRNA [55]. In this method, in situ indexing is used to get the position of each random barcode. This platform is much better than Visium as its resolution is up to 10 μm with improved sensitivity as it can analyze 500 transcripts per bead [56]. Another platform named high-definition spatial transcriptomics (HDST) further improved the resolution power such as Slide-Seq by using the beads instead of glass slides which were deposited in wells [57]. In addition, the DBiT-Seq [58] platform has been innovated with microfluidics channels which are used to apply polyT barcodes for capturing RNA in tissues. Stereo-seq is another useful platform that involves the use of randomly barcoded DNA nanoballs that obtain RNA analysis up to nanoscale resolution [59]. Seq-Scope performs spatial mapping of RNA transcripts within the nucleus and cytoplasm using spatial barcoding at a subcellular resolution [60]. A polony-derived gel oligo array was used in Pixel-Seq to capture RNA analytes leading to an improved resolution of ~200-fold as compared with the existing methods [61]. 

### 4.2. Imaging-Based Approaches

There are two main types of imaging-based approaches that are used for spatial transcriptomics: in situ hybridization and in situ sequencing-based methods. 

#### 4.2.1. Multiplex Error Robust Fluorescent In Situ Hybridization (MERFISH)

MERFISH (multiplexed error-resistant fluorescence in situ hybridization) can test thousands of RNA molecules in cancer tissue samples. MERFISH uses a two-step approach for the analysis of RNA molecules from cells [62]. The method consists of hybridization of specific encoding probes composed of complementary sequences and two flanking readout sequences. The approach detects the analyte through several rounds of hybridization within 15min compared with contemporary methods which required over 10 h for the analysis of mRNA molecules. Despite several advantages, the technique faced two main hurdles in its widescale applicability in the form of reduced sensitivity, and thus required an increase in the round of hybridizations of probes for transcripts analysis [62]. However, this deficiency was overcome by the development of a new modified method named Hamming distance error correction code. This approach involved the removal of fluorophores with a chemical reaction and the ability to image multicolor barcodes resulted in a decrease in hybridization rounds that led to improving the sensitivity of MERFISH [63].

Single-molecule fluorescence in situ hybridization (smFISH) offers both quantitative measurements of RNA expression and even RNA spatial localization by directly imaging individual RNA molecules in single cells [64,65]. The smFISH technique is considered useful for understanding a vital biological phenomenon from cell division to body patterning in the development process. Furthermore, recent progress in multiplexed smFISH [62,66,67] and in situ sequencing [62,68] allows an increase in detection limit and sensitivity of RNA entities within tissue cells showing a potential of profiling hundreds to thousands of RNAs at a single-cell resolution [62,69,70]. By these methods, it has been possible now to visualize the internal organization of transcriptome, thus enabling a better understanding of cell diversity based on the RNA expression profiles [71]. Among these approaches, MERFISH multiplexes with smFISH use specific barcodes (error-resistant barcodes) for each RNA species, and thus each RNA molecule is labeled with an oligonucleotide having a unique barcode. Imaging of these adjacent labeled RNA molecules with smFISH permits analysis of transcriptome at a single-cell level [62].

It would be worth mentioning here that in the MERFISH transcriptome analysis method, the analysis of RNA molecules is based on a fluorescent signal generated by fluorescently labeled probes tagged to RNA molecules. The intensity of the signal is usually high enough for the detection of RNA molecules, however, reduced signal intensity renders low detection and may lead to misinterpretation of biological processes [63,72]. Low signal intensity is recorded by the application of cameras with extended exposures from high-power laser illuminations. Thus, signal brightness is improved by laser illuminations. The length of the RNA molecule also matters for the binding of multiple probes; smaller RNA molecules allow a limited number of probes binding and thus reduce signal emission, leading to reduced RNA sensitivity with the MERIFISH technique [72,73].

#### 4.2.2. Fourth-Generation RNA-Seq

Advances in the Fourth-generation RNA-seq platforms, especially in situ sequencing (ISS) and fluorescent ISS (FISSEQ), have greatly helped to achieve the objectives and aims of RNA-seq techniques [69,74]. ISS technique is highly sensitive and involves the use of padlock probes and rolling circle amplification (RCA) to generate targeted sequencing libraries, which are later sequenced by the application of NGS techniques. The uniqueness of the ISS approach is that it can sequence up to 256 RNA transcripts during a single round of hybridization and thus possess the potential of detection of a combatively higher number of transcripts during tissue analysis experiments. On the other hand, the fluorescent in situ sequencing (FISSEQ) uses random hexamers and a sequencing primer tag for stating the RT reaction [75]. Once the reaction is completed, products are sequenced by the application of NGS sequencing techniques as used for ISS. The FISSEQ method is better than the ISS methods as it can create random libraries, perform unbiased analysis, and detect low RNA copy numbers during the analysis of RNA from single cells [76]. However, ISS sensitivity is two times greater than FISSEQ methods [77]. Despite several unique features of ISS and FISSEQ techniques in terms of their detection limits, both techniques are in their early stage of development and need immense improvements in sample preparation, increasing efficiency, improvement in computational techniques, and imaging scale. 

#### 4.2.3. Laser Capture Micro-Dissected RNA-Seq

Laser capture micro-dissected RNA-seq (LCM-RNAseq) is an advanced approach to overcome the limitations faced by bulk RNA-seq techniques [78]. The LCM-RNAseq approach involves the laser capture dissection of cells and then RNA-seq. Over the past few years, advances in LCM-RNAseq methods have enabled researchers to quantify low-input degraded RNA molecules extracted from FFPE tissues. For instance, Singh et al. [79] studied tumor tissue heterogeneity by acquiring sequencing data from 10 LCM isolated single cells. After this study, another LCM-Smart3seq technique was used which was capable of performing analysis of a very low quantity of RNA analytes that were obtained from cells processed by LCM [80]. Later, further advancement in LCM-RNAseq techniques such as FFPEcap-seq helped the researchers to accurately detect/quantify RNA from FFPE tissues. This method consisted of the capping of 5′ ends of RNA molecules extracted from FFPE tissues [81]. The first study that involved the use of LCM-RNAseq on cancer tissues was tested on lung cancer [82]. This work enabled the researchers to assess the genes which were involved in the growth of lung cancer. Over the last few years, LCM-RNAseq has revealed the spatial organization of various types of cells in cancer tissues [83]. The analysis of tissue sections obtained from human glioblastoma (BGM) by the LCM-RNAseq technique showed the presence of an array of interconnected channels with a specific micro-environment that was responsible for the proliferation and migration of cancer cells through the generation of specific signals [84,85]. Some of the advantages and disadvantages of RNA-Seq techniques are presented in Table 2.

## 5. Integration of Single-Cell RNA-Seq with Spatial Mapping Techniques

Single-cell RNA-seq techniques can generate the sequence information from slides while preserving the histological context of the area of interest [86] and identifying thousands of genes in the sampled tissues. Besides, single-cell RNA-seq techniques provide the clinical tools which can contribute towards the early detection of cancer for the development of suitable treatment strategies [87]. In addition, these techniques can also help to identify the location of mutated/aggressive clones inside the tumor, distinguish between its center and infiltrating edges, provide an evaluation of changes at the molecular level in the stroma located inside and outside the tumor, and help with the detection of epithelial-mesenchymal transition [88]. Over the last few years, it has been demonstrated that single-cell RNA-seq techniques can sequence RNA within the tumor cell from both fresh-frozen cells or fixed tissues without compromising their spatial context [89]. It can also be used to incorporate the spatial features directly to specific genetic elements in organoid specimens or native tissue using image analysis approaches. It can also characterize and map the sequences obtained from the tumor tissue with the spatial distribution of the novel microenvironment that includes the complex arrangement of different types of cells which are regulated by the interplay of single cells [90]. Thus, the main purpose of spatial mapping techniques is to get the whole genome/transcriptome data of all cells on the full slide and use it for further improvement using sequencing techniques [91].

Single-cell RNA-seq has been combined with other omics technologies for better analysis and understanding of the RNA analytes in their spatial context. In situ visualization of transcriptome by a single-cell RNA-seq gives multiplexed information for the expression of genes, cell types, and disease progression patterns. On the other hand, the in situ sequencing (ISS) technique, as mentioned earlier, by using padlock probes and amplification by a rolling circle amplification, spatially displays expressions of RNA in a range of tissue sections. The study reported by Gyllborg et al. used hybridization-based ISS (HybISS) for understanding the spatial localization of RNA in human and mouse brain tissue samples [92]. This probe’s design modifications in the HybISS platform lead to improved combinatorial barcoding with an improved detection limit of transcripts, and thus portrayed localization of RNA transcripts in a spatial context. In another study, Asp et al. combined different platforms such as single-cell RNA-seq with spatial transcriptomics and ISS platforms to reveal the cellular architecture of the human heart in different developmental stages [93]. Integration of spatial transcriptomics and single-cell RNA-seq allowed the mapping of various types in developing human hearts while ISS allowed the understanding of subcellular details in their physical locations. Moncada et al. combined single-cell RNA-seq with microarray-based spatial transcriptomics and found variations in gene expression patterns in a spatial manner in pancreatic ductal adenocarcinomas. This conformed with the spatial localization of various types of cells and subsets including cancer cells, macrophages, ductal cells, and dendritic cells in pancreatic tumors [53]. Another study was carried out to visualize the spatial localization and cellular heterogeneity of pancreatic ductal adenocarcinomas through geographical positional sequencing (Geo-seq) that combined both laser capture micro-dissected (LCM) and single-cell RNA-seq techniques [94]. This technique allowed to differentiate various types of subpopulations of ductal macrophages, cells, dendritic cells, and cancer cells in a spatial manner. In another study, the spatial distribution of RNA analytes from a tissue section was performed using an array of transcription primers fixed on the glass slide [7]. The tissue sections were mounted on a glass slide, which was followed by the permeabilization step and reverse transcription on glass slides to synthesize cDNA. Thus, this technique involved the analysis of RNA isolated from tissue sections fixed on an array of transcription primers with unique positional barcodes over the glass slide and the construction of their spatial distribution in a two-dimensional map. Single-cell RNA-seq techniques concerning spatial profiling have been described precisely in a recently published review article, where authors have documented the further details in advances of the state of art in the context of the analysis of liver histology [95]. In the context of the spatial profiling of transcriptome of cancer tissue, Ji et al. used single-cell RNA-seq combined with spatial transcriptomics and multiplexed ion beam imaging techniques to analyze the spatial localization of tumor-specific keratinocytes (TSK) within breast carcinoma [54]. The technique enabled the team to visualize the existence of a population of TSL, immune infiltrates displaying cellular heterogeneity along edges of the tumor.

Single-cell RNA-seq studies identified two cancer cell populations for the pancreatic ductal adenocarcinoma (PDAC-A) tumor that seemed to be quite different from each other histologically [96]. There was enrichment of endothelial cells, monocytes, and fibroblasts in the PDAC-A sub-region 1 but a substantial deficiency of cancer cells. Figure 4 shows some of the recently used approaches integrated with spatial transcriptomics for the analysis of cancer tissues.

## 6. Clinical Applications of Single-Cell RNA-Seq Techniques

Single-cell RNA-seq is a well-known technique for clinicians and it is expected that it will be ready soon for clinical applications. Up till now, the transcriptome of tissue samples of various disease origins obtained from humans has been mapped with the help of single-cell RNA-seq. The target of most single-cell RNA-seq studies has been to map the transcriptome of tissues to explore the molecular mechanism of disease progression. This is deemed very useful for differentiating cellular subpopulations stratifying various types of disease categories as well as assessing different therapeutic responses. In cancers, calculations of transcripts copy number, gene mutations, and modifications at a single-cell resolution are used for determining tumor heterogeneity, clonal lineages, and metastatic variations. For this application, target gene panels for single-cell RNA seq can perform profiling of target RNA molecules with improved efficacy. 

Tumor heterogeneity is one of the main features of cancers and plays a major role in drug resistance. The infiltration of stroma and the presence of diverse types of cells in tumor tissues largely compromise the results obtained by bulk RNA-seq. However, single-cell RNA-seq techniques, by exploring the intratumoral transcriptomic heterogeneity of cancers [69], play a significant contribution in evaluating therapeutic responses. In this context, a group of researchers found that drug-resistant breast cancer cells contain RNA variations in genes that are involved in microtubule organization, cell adhesion, and cell surface signaling [97]. However, there were no drug-tolerant-specific RNA variants present in stressed cells or untreated cells. It was concluded that the formation of RNA variants is associated with tumor heterogeneity and thus drug resistance in cancer cells compared with normal cells. Single-cell RNA-seq analysis offers a useful opportunity for tumor treatment. It is seen that a targeted therapy eliminates a specific clone of cells while other cell populations remain unharmed due to intratumoral heterogeneity. This challenge was overcome by the development of therapeutic strategies, which were able to target multiple tumor subpopulations. Kim et al. analyzed the drug target pathways which have a key role in drug resistance in metastatic renal-cell carcinoma using single-cell RNA-seq. Then, they used combinatorial therapeutic strategies that display improved response in vitro and in vivo as compared with the monotherapy strategy [98].

Single-cell RNA-seq is applied to identify specific gene signatures for a cell type and the characterization of known and unknown cell types as well as subtypes within and surrounding tumors [99,100,101]. This has resulted in the improvement in cancer treatment outcomes by exploring unknown complex pathways involved in heterogeneous tumor tissues. It is well known that T cell infiltration and its characteristics have important relationships with prognostic outcomes [102] which were used for their clinical responses in cancers. This was used by Zheng et al. [100], who found that infiltrating lymphocytes in liver cancer has a distinctive functional composition of T cells in the hepatocellular carcinoma (HCC). The team was able to identify 11 large subsets and also found specific subpopulations at a single-cell level which offer useful targets for clinical applications [100].

## 7. Existing Challenges and Prospects

Single-cell RNA-seq technique and spatial transcriptomics are valuable platforms for the identification of cancer cells in their physical locations, which can be used for exploring intra-tumor heterogeneity and epigenetic modifications for individualized therapeutic approaches. This can have huge effects in clinical applications, but their practical application is restricted by different limitations. Firstly, there is comparatively high experimental time and costs required for their use in common clinical labs and thus it is highly recommended that their costs and processing time should be minimized. There is a dire need for the development of platforms that can perform analysis of formalin-fixed paraffin-embedded tissues, as in routine practice in the clinics most of the samples are preserved in formalin-fixed and are paraffin-embedded [103,104]. These advances will be able to encourage the prognosis and diagnosis of cancer tissues with single-cell RNA-seq and spatial transcriptomics which will improve treatment interventions of cancers and eventually enhance cancer patients’ survival.

One of the main challenges in the single-cell RNA-seq technique is that it requires the isolation of single cells by complex sorting techniques, which may compromise the structure of cells resulting in unknown transcription changes during the analysis of cells. This challenge is overcome using cell lines and organoids for single-cell RNA-seq analysis, but these cannot be alternative to tissue samples as these lack intricate interactions that exist between cancer cells and their microenvironment. 

Most RNA-seq methods depend on poly(A) tail capture to enrich mRNA and deplete abundant and uninformative rRNA. Thus, these methods are often restricted to sequencing polyadenylated mRNA molecules. However, recent studies are now starting to appreciate the importance of non-poly(A) RNA, such as long-noncoding RNA and microRNAs in gene expression regulation. This has led to the development of a small-seq technique which is a single-cell method that can capture small RNAs (<300 nucleotides) such as microRNAs, fragments of tRNAs, and small nucleolar RNAs in mammalian cells [105]. This method uses a combination of “oligonucleotide masks” (that inhibit the capture of highly abundant 5.8S rRNA molecules) and size selection to exclude large RNA species such as other highly abundant rRNA molecules. To target larger non-poly(A) RNAs, such as long non-coding mRNA, histone mRNA, circular RNA, and enhancer RNA, size selection is not applicable for depleting the highly abundant ribosomal RNA molecules (18S and 28s rRNA) [106]. Single-cell RamDA-Seq is a method that achieves this by performing reverse transcription with random priming (random displacement amplification) in the presence of “not so random” (NSR) primers specifically designed to avoid priming on rRNA molecules [107]. While this method successfully captures full-length total RNA transcripts for sequencing and detects a variety of non-poly(A) RNAs with high sensitivity, it has some limitations. The NSR primers were carefully designed according to rRNA sequences in the specific organism (mouse), and designing new primer sets for other species would take considerable effort. Recently, a CRISPR-based method named scDASH (single-cell depletion of abundant sequences by hybridization) demonstrated another approach to depleting rRNA sequences from single-cell total RNA-seq libraries [108]. 

In the single-cell RNA-seq sequencing technique, the quality and quantity of RNA isolated and amplified play an important role in the precise analysis of RNA transcripts. The mapping of RNA molecules present in a cell is a random variable that depends upon the number of RNA present in a cell. This is largely affected by reverse transcription reaction, the extent of cDNA amplification, and the efficacy of RNA capture from the cells. This can lead to an obscure relationship between an actual number of reads and counted genes. This challenge is overcome by the use of unique molecular identifiers (UMIs) which help to prevent differences in amplified cDNA molecules and thus provide a useful way for estimating the accuracy of reaction by estimating the number of captured and reverse-transcribed mRNA molecules [109,110]. 

LCM-RNAseq is a useful technique for the analysis of transcriptome from fresh and FFPE tissue samples, but RNA extracted from FFPE tissue samples results in the extraction of low quantity and quality of mRNA. The analysis of this low-quality RNA with LCM-RNAseq methods affects the integrity of RNA. It is, therefore, imperative to improve LCM-RNAseq analysis techniques to increase the quality and stability of RNA biomolecules. This is achieved by using an LCM instrument with a suitable IR laser and the use of an appropriate RNA-seq library preparation kit which prevents RNA degradation [78]. In this context, the use of SMARTer Stranded Total RNA-Seq Kit v2 is considered beneficial for extraction of very low concentrations (i.e., 250 pg-10 ng) of RNA from FFPE tissue samples. Furthermore, additional rounds of PCR cycles are performed to increase the output of cDNA from low-quality RNA transcripts. There is another challenge that is encountered for the analysis of RNA by the LCM-RNAseq technique as this method is time-consuming and the technique is limited to analyzing RNA from 10 to 100 cells only. It is observed that bulk RNA-seq can sequence RNA from >1 × 10^6^ cells. The limitation of LCM-RNAseq has been addressed now using alternative techniques or improved technological platforms. For instance, He et al. developed a new algorithm named (ADVOCATE) which is capable of analyzing the data derived by LCM-RNAseq to obtain expression profiles of different genes in the tissue sections of pancreatic ductal adenocarcinoma (PDA) [104].

Mapping of RNA across tissue sections was originally limited for snap-frozen mammalian tissues and fresh plant tissues [111]. However, with the advent of novel spatial transcriptomics techniques, now a few studies have been performed on paraffin-embedded and formalin-fixed samples with modification in analytical platforms to locate expression of RNA in the spatial context in biological tissues [112]. Among a few initial studies, the spatial transcriptome analysis was carried out on locating mRNA expression in prostate cancer [113] and melanoma. After the analysis of 6750 regions in the prostate and 2200 sites in melanoma, it was concluded that there exists an extreme heterogeneity in distinct gene expression signatures and coexistence of unique expression profiles of different sets of genes in tissue biopsies [52]. Yoosuf et al. designed a study to investigate the difference between invasive ductal carcinoma (IDC) and ductal carcinoma in situ (DCIS) using spatial transcriptomics datasets of breast cancer with a machine learning technique [114]. The team was able to find a prediction accuracy of 91% for IDC and 95% for DCIS from the application of spatial transcriptomics signatures and training the machine learning methods. One of the main advantages of spatial gene localization by transcriptomics technique is that it did not need pre-knowledge of gene sequences or specific equipment and its outcome is much higher compared with other allied tissue analysis methods. 

One of the main challenges faced in profiling RNA transcripts in their spatial context was that molecular techniques were unable to analyze mRNA expression up to the single-cell level due to limitations of microarray spot size and spacing. The advances made by Stahl’s group in 2019 were able to visualize gene expression with improved spatial resolution up to 1400×. This can also detect spatial gene expression patterns at the single-cell level [7]. This led to an improved understanding of cancer biology, especially tumor heterogeneity and therapeutic outcomes. However, techniques invented by Stahl et al. have limited resolution compared with the captured mRNA data. The improvement brought about in the 10x genomics chromium platform has enabled the profiling of up to 10,000 cells in a single experiment, however, it is only possible to analyze only a few thousand mRNA molecules from a single cell. This issue can be addressed up to a certain degree by deeper sequencing, however, even this attempt will be far less for the analysis of the full transcriptome of a single cell. In the case of bulk RNA-seq, LCM-RNAseq, and single-cell RNA-seq, there is one common challenge, which is the loss of spatial information of RNA due to cell dissociation or the micro-dissection step involved at an early stage of the protocol, resulting in a reduced understanding of cell function and thus pathological changes [7]. 

The pathological nature of cancer tissues is decided based on the single-cell transcriptomic profile differences of healthy and cancer patients. However, we lack standard data regarding disease subtypes, duration, severities, and drug response outcomes for clinical considerations. One of the major hurdles of single-cell RNA-seq is that we still lack a definition of this platform for clinical application and monitoring responses to drugs at the single-cell level. Recent progress in the integration of single-cell RNA-seq with spatial transcriptomics will immensely help in exploring undiscovered biomarkers and agents involved in tumor development and will pave a path for better therapeutic outcomes.

## 8. Conclusions

Single-cell RNA-seq techniques have been very useful for performing transcriptome analysis of single cells within and across cancerous tissues. The techniques applied in recent years are mostly focused on the isolation of cancer cells from cancer tissues, extraction of RNA, and their analysis by NGS techniques. Few studies have attempted to visualize RNA expression in their native tissue context in combination with other techniques over the last few years, but the interest in profiling transcriptome at a single-cell level in its spatial context is growing every day as evident from the number of published works. Nevertheless, recent developments in single-cell RNA-seq techniques with other allied approaches have a huge impact on exploring the tumor heterogeneity that will have a profound influence on finding the personalized medicine to prevent cancer relapse, which is a major issue in oncology. Moreover, in the field of translational research, single-cell RNA-seq possesses a huge potential in determining single-nucleotide variation (SNV), methylation patterns, copy number variation (CNV), microsatellite instability, and gene rearrangements/translocations for understanding tumor heterogeneity of cancer tissues for use in clinical settings.

## Figures and Tables

**Figure 1 ijms-23-03042-f001:**
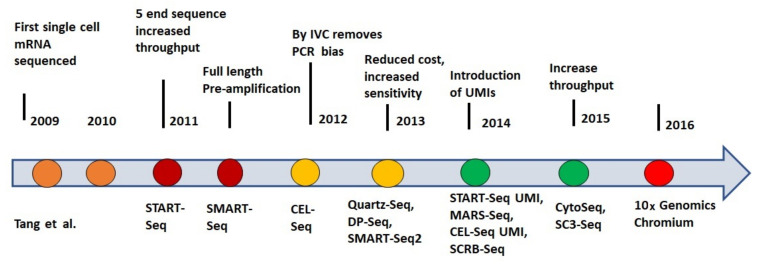
Presents a timeline of the discovery of RNA sequencing techniques and their improvements in efficiency and sensitivity with innovations in techniques [9].

**Figure 2 ijms-23-03042-f002:**
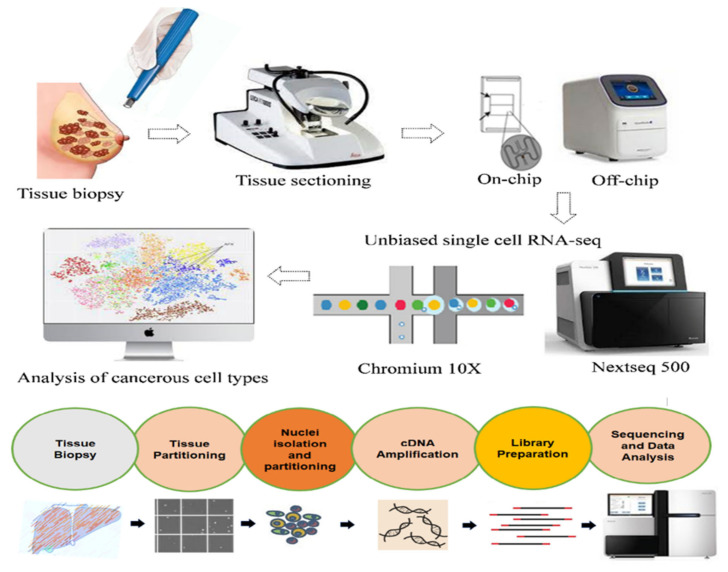
A schematic of various steps used for the analysis of biopsy tissue samples by RNA-seq techniques such as isolation and sequencing of single cells, preparation of RNA library, and single-cell level transcriptome analysis.

**Figure 3 ijms-23-03042-f003:**
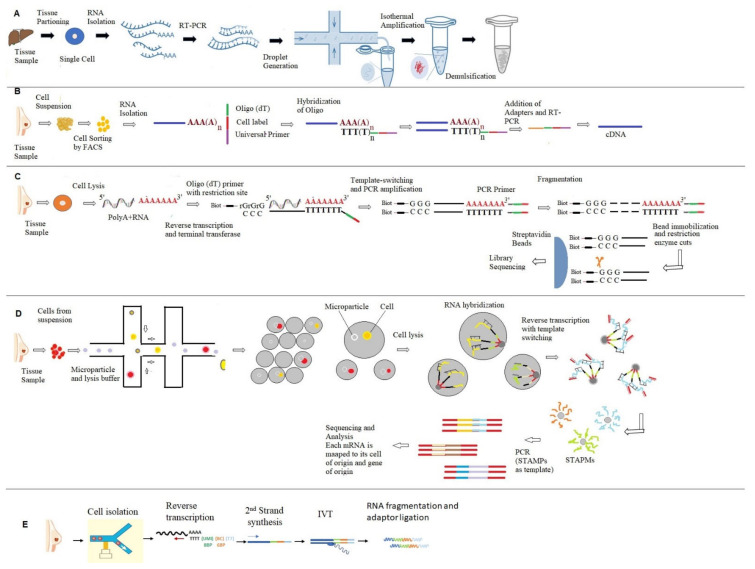
Shows different types of RNA-seq techniques used for the analysis of RNA from cancer tissues. (**A**) depicts cell expression by linear amplification and sequencing method; (**B**) displays single-cell RNA barcoding and sequencing (SCRB-seq) approach; (**C**) displays steps involved in switching mechanism at the end of the 5′-end of the RNA transcript sequencing (Smart-seq2), (**D**) represents various steps used for the analysis of transcripts by Drop-sequencing (Drop-seq), and (**E**) shows various steps involved in the Massively Parallel RNA Single-Cell Sequencing Framework (MARS-seq).

**Figure 4 ijms-23-03042-f004:**
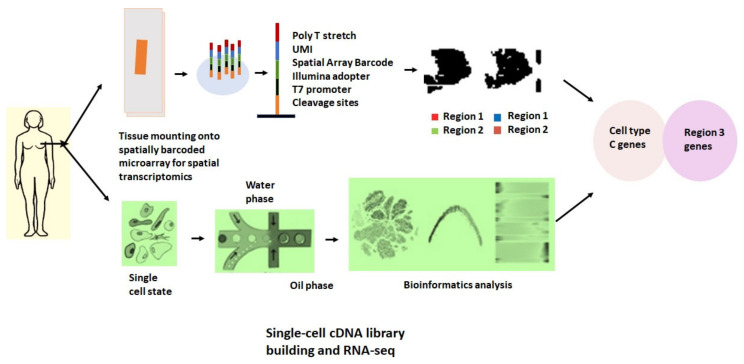
Single-cell RNA-seq (scRNA-seq) helps in dealing with solid and circulating tumor tissues in cancer research. The figure shows an analysis of tissue samples taken from cancer patients by mounting them on glass slides and then tissue permeabilization on glass slides. RNA is amplified using UMIs and imaged without losing the spatial localization of RNA analytes. In the above figure, the second route shows the isolation of cells from tissue samples up to single-cell level, then cell sorting by microfluid device, and then clustering of cells according to RNA sequences performed with NGS.

**Table 1 ijms-23-03042-t001:** Comparison of single-cell techniques in the form of the methodology used and advantages gained for analysis of mRNA analytes.

Technique	UMI	mRNA Priming	cDNA Preamplification	Library Generation	Transcript Coverage	Strand Specificity	Positional Bias	Costs	Reference
CEL-seq2	Yes	Poly T	In vitro transcription	Transposon tagmentation	3′-only	No	Weakley 3′	High	[30]
SCRB-seq	Yes	Poly T	PCR	RNA fragmentation and adapter ligation	Nearly full length	No	Strongly 3′	High	[9]
Smart-Seq	No	Poly T	PCR	Transposon tagmentation	Full length	No	Medium 3′	High	[38]
Drop-seq	Yes	Poly T	PCR	Transposon tagmentation	3′-only	Yes	3′ only	Low	[41]
MARS-seq	Yes	Poly T	In vitro transcription	RNA fragmentation and adapter ligation	3′-only	Yes	3′ only	Low	[14]
10×Genomics	Yes	Poly T	PCR	cDNA fragmentation, adapter ligation, and library amp	3′-only	Yes	3′ only	Low	[42]

**Table 2 ijms-23-03042-t002:** Advantages and limitations of RNA-Seq techniques for spatial mapping of biomarkers used in clinical oncology.

Type	Strength	Weaknesses	Suitable Applications
Bulk RNA-seq	Well-developed, cost-effective, and high throughput technique	Unable to determine spatial content; gene expression profiling is average	Whole transcriptome-based biomarker discovery, targeted RNA-seq panel for gene fusion
MERFISH	High-throughput, high-sensitivity, high-multiplex power	Reduced specificity and off-target binding	Spatial organization of the transcriptome inside the cells, 3D organization of the chromatin and chromosome, spatial atlases of cells in complex tissues
LCM-RNAseq	Performs cell-specific gene expression analysis	Low-quality data, time-consuming, unable to perform spatial profiling	Applied for tumor heterogeneity to the specific population of cells
Single-cell RNA-Seq	Capable to perform >10,000 single-cell gene expression analysis	Applicable to a limited number of unique transcripts, unable to reveal spatial content, high cost	Characterization and discovery of cell type tumor heterogeneity
Digital Spatial Profiling	Useful for FFPE materials, spatial profiling	Unable to reveal sequence information, restricted to a small number of gene panels only	Biomarker discovery, tumor microenvironments
Spatial transcriptomics	Spatial profiling, whole transcriptome analysis, sequence information	Time-consuming, the early phase of development	Tumor microenvironments, tumor heterogeneity
Fourth-generation RNA-seq	Potential of in situ sequencing	Not properly well developed	Great future potential but not demonstrated yet

## Data Availability

This is a review article and does not contain any raw data.

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
