# Peer review of "Single-Cell RNA Sequencing with Spatial Transcriptomics of Cancer Tissues"

_ijms, 2022, doi:10.3390/ijms23063042_

Round 1

Reviewer 1 Report

Ahmed et al. reviewed single cell RNA sequencing thoroughly. This review is well written, and educational for readers of IJMS. Figures and schemas are impressive. I have a few minor comments.

1) I would like to know more clinical application of single cell RNA analysis. How does the clinicians/oncologists apply this technology to cancer therapeutics?

2) Please revise the reference section.

Author Response

Response to the Reviewer #1

Reviewer #1: Ahmed et al. reviewed single-cell RNA sequencing thoroughly. This review is well-written and educational for readers of IJMS. Figures and schemas are impressive. I have a few minor comments.

Reviewer 1: 1.  I would like to know more clinical applications of single-cell RNA analysis. How do the clinicians/oncologists apply this technology to cancer therapeutics?

Authors Response:

We thank the reviewer for thoroughly reviewing our manuscript and proving us with detailed comments, which helped us a lot to improve the quality of it.  

We have added clinical applications of single-cell RNA analysis in the revised draft which can be read on page 14, lines 540-579 as

  1. Clinical applications of single-cell RNA-seq techniques

Single-cell RNA‐seq is a well-known technique for clinicians and it is expected that it will be ready doon for clinical applications. Up till now, transcriptome of tissue samples of various disease origins obtained from humans have been mapped with the help of single-cell RNA‐seq. The target of most of single-cell RNA-seq studies have been to map transcriptome of tissues to explore molecular mechanism of disease progression. This is deemed very useful for differentiating cellular subpopulations stratify various types of disease categories as well as to assess different therapeutic responses. In cancers, calculations of transcripts copy number, gene mutations and modifications at a single cell resolution is used for determining tumor heterogeneity, clonal lineages, and metastatic variations.  For this application, target gene panels for single-cell RNA seq that can perform profiling of target RNA molecules with improved efficacy.

Tumor heterogeneity is one of main feature of cancers and play major role in drug resistance. The infiltration of stroma and presence of diverse types of cells in tumor tissues largely compromises the results obtained by bulk RNA-seq. However, single-cell RNA-seq techniques by exploring the intratumoral transcriptomic heterogeneity of cancers [69] play a significant contribution for evaluating the therapeutic responses. In this context, a group of researchers found that drug resistant breast cancer cells contain RNA variations in genes that are involved in microtubule organization, cell adhesion and cell surface signaling [97]. However, there were no drug-tolerant-specific RNA variants present in stressed cells or untreated cells. It was concluded that formation of RNA variants is associated with tumor heterogeneity and thus drug resistance in cancer cells compared to normal cells. Single-cell RNA-seq analysis offers useful opportunity for tumor treatment. It is seen that a targeted therapy eliminates a specific clone of cells while other cell populations remain unharmed due to the intratumoral heterogeneity. This challenge was over-come by development of therapeutic strategies which was able to target multiple tumor subpopulations. Kim et al. analyzed the drug target pathways which have a key role in drug resistance in metastatic renal cell carcinoma using single-cell RNA-seq. Then they used a combinatorial therapeutic strategy that display improved response in vitro and in vivo as compared to monotherapy strategy [98].

Single-cell RNA-seq is applied to identify specific gene signatures for a cell type and for the characterization of known and unknown cell types as well as subtypes within and surrounding tumors [99-101]. This has resulted in the improvement in cancer treatment outcomes by exploring unknown complex pathways involved in heterogeneous tumor tis-sues. It is well known that T cell infiltration and their characteristics have important relation with prognostic outcomes [102] which was used for their clinical responses in cancers. This was used by Zheng et al. [100] and found that infiltrating lymphocytes in liver cancer have distinctive functional composition of T cells in the hepatocellular carcinoma (HCC).  The team was able to identify 11 large subsets and also found specific subpopulations at single-cell level which offer useful targets for clinical applications [100].

Reviewer 1: 2. 

(1)     Please revise the reference section.

Authors Response:

We have revised all the references and modified them according to text which can be seen on page 17-23.

Reviewer 2 Report

Single-cell RNA sequencing progress towards spatial transcriptomics: state of art and prospects - Review

In this review by Ahmed et al., the aim of the authors is to provide an overview of single-cell RNA-seq techniques currently used for analysis of cancer tissues, focusing specifically on deciphering intratumor heterogeneity. While the topic is relevant to current research, the manuscript needs a deep reshape in terms of clarity, organization and presentation. Please see below.

General comments

- One of the major concerns of this manuscript is the lack of clarity and the poor level of English. Moreover, despite that ideas may be relevant, they are not clearly and coherently organized. Some sentences are not very precise; they tend to be too general. The reader is lost and confused. An expert should do correction and editing of the writing.

- The pros and cons are just listed and not discussed (see below).

- The recent advances in single-cell RNA-seq techniques are mentioned, but not detailed.

- The main novelty of this review article is stated to be the following: “it presents an integration of single-cell RNA-seq with data generated by omics technologies for a more comprehensive analysis of intratumor heterogeneity of cancer tissue samples.” However, this part is only dealt with in the last section of this review article, and is not that great and not well presented (see below).

- References are missing.

- Some references are present more than once in the references section. Eg. “K.H. Chen, A.N. Boettiger, J.R. Moffitt, S. Wang, X. Zhuang, RNA imaging. Spatially resolved, highly multiplexed RNA profiling in single cells, Science, 348 (2015) aaa6090, doi: 10.1126/science.aaa6090.” appears five times.

- Abbreviations are not always given at the right place. For e.g. LCM first appears on Page 9 while its full form is given on Page 12.

Specific comments

Introduction

- Please re-organize and clarify the ideas presented

Figure 1

- The general experimental outline of scRNA-seq is not described in the text. The sentence referring to Figure 1 at the end of the introduction seems at the wrong place.

- Copy number calculations are not illustrated in Figure 1 as stated in the legend.

History of Single-cell RNA-seq Techniques

- 10X Genomics is not mentioned, though it is a very frequently used scRNA-seq technology.

- Technological advances in the microfluidic and nanotechnologies are not discussed. They had a great impact in the field. They allowed higher-throughput profiling in the same cell, with large numbers of cells simultaneously and with dramatically reduced costs.

- This section should be written again to better put forward the evolution of the techniques and the reasons why they evolve in a first part, and then the advancements that resulted due to the evolution of the techniques.

Figure 2

- Figure 2a:

  • Microfluidic-based techniques (10X Genomics, Drop-seq and inDrop) are not present on timeline.
  • Do the colored circles mean something?

-Figure 2b: Similar to Figure 1, and therefore can be merged to Figure 1.

Methods of Single-cell RNA-seq Techniques

- General structure is not constant for all techniques presented.

- For some techniques, the general sentence(s) is too long or repetitive.

- The first study using or developing the technique is not always mentioned. References are missing. For e.g., only the reference for the CEL-seq article is given, not the one for CEL-seq2.

- A brief description of the technique is not always provided in the text, although Figure 3 shows the main steps.

- The pros are presented as a list and remain very vague. For e.g., for the Drop-seq technique, the following sentence is not precise and not specific to the technique: “The merits include evaluation of single-cell sequences of similar patterns, identification of gene-specific mRNA strands via single molecular and cell barcodes” Moreover, the pros are not discussed.

- For some techniques, the cons are not mentioned (for e.g. not given for the CEL-seq2 or SCRB-seq technique). When they are, they are just listed and not discussed.

- The pros and cons of the different techniques are only briefly compared. A table comparing the methodology as well as the advantages and disadvantages of each technique presented could be interesting.

- Among the microfluidic-based techniques, only the Drop-seq technique is presented. 10X Genomics has gained popularity over the last years. Moreover, it is mentioned in the “Spatial transcriptomics” section as having played a major contribution in the development of spatial analysis.

- Sentences are sometimes very vague. For e.g., in the sentence “Moreover, improvements in CEL-seq2 protocols”, the precise improvements are not mentioned.

- The SMART-seq and SMART-seq2 sections could be merged into one section. The pros and cons of the SMART-seq technique are not presented. The cons would help understand why and how the technique was improved to lead to SMART-seq2.

Figure 3

- The sentence introducing Figure 3 should be placed at the end of the introductory paragraph of the “Methods of Single-cell RNA-seq Techniques” section.

- Keep the same illustrations and colors from A-D. For e.g., the same tissue sample, the same color for the cell suspension, … This will ease comparison between techniques.

- No scheme is given for MARS-seq while it is given for the other four techniques.

Spatial transcriptomics

- Spatial transcriptomics: gene expression in different locations in a tissue but also in different subcompartments within a cell.

- Main groups of methodology for spatial transcriptomics are not presented and described as such in this section. Some are dispersed throughout this section, intermingled with results obtained using these techniques. It would be better to separate the description of the methodologies, their biological applications and the results already obtained using this technique.

- Some details on the methodologies are in the next section “Integration of single-cell RNA-seq with spatial mapping techniques”. I think it would be better for them to be in this section. Separate the different methodologies, that is, a subsection for MERFISH, another for smFISH. Compare them in the text. Table 1 gives a good overview of the pros and cons of the methodologies presented. It could be completed with the other methodologies not mentioned (e.g. MERFISH).

Integration of single-cell RNA-seq with spatial mapping techniques

- The different ways of integrating the scRNA-seq and spatial mapping techniques are not clearly separated from one another. The methodology is not separated from the results obtained using the technique.

- A detailed description of one study is given. Why it is so? The reference of the study is not given. It could be more discussed.

- This section is quite poor in terms of data.

Figure 4

- Figure 4 shows two (not some) approaches integrated with spatial transcriptomics for the analysis of cancer tissues.

- Legend: In the above figure, the second route shows the isolation of barcoded cells from tissue samples …

Existing challenges and prospects

- The first paragraph is not related to existing challenges and prospects. It gives a summary of the biological applications of scRNA-seq and scRNA-seq coupled to spatial data. The ideas in this paragraph can be highlighted in previous sections instead.

- The challenges and prospects are not clearly separated.

Author Response

We thank the editorial team and all reviewers for their constructive feedback on our manuscript.  We have addressed all the comments and we believe that our manuscript is now significantly improved. We have addressed each reviewer’s and editor’s comments below and in red font in the revised manuscript.

Round 2

Reviewer 2 Report

The authors have addressed my concerns and the paper is much clearer and well organized.